# Population Pharmacokinetics and Dose Optimization Based on Renal Function of Rivaroxaban in Thai Patients with Non-Valvular Atrial Fibrillation

**DOI:** 10.3390/pharmaceutics14081744

**Published:** 2022-08-21

**Authors:** Noppaket Singkham, Arintaya Phrommintikul, Phongsathon Pacharasupa, Lalita Norasetthada, Siriluck Gunaparn, Narawudt Prasertwitayakij, Wanwarang Wongcharoen, Baralee Punyawudho

**Affiliations:** 1Division of Clinical Pharmacy, Department of Pharmaceutical Care, School of Pharmaceutical Sciences, University of Phayao, Phayao 56000, Thailand; 2Unit of Excellence on Pharmacogenomic Pharmacokinetic and Pharmacotherapeutic Researches (UPPER), School of Pharmaceutical Sciences, University of Phayao, Phayao 56000, Thailand; 3Division of Cardiology, Department of Internal Medicine, Faculty of Medicine, Chiang Mai University, Chiang Mai 50200, Thailand; 4Department of Pharmaceutical Care, Faculty of Pharmacy, Chiang Mai University, Chiang Mai 50200, Thailand

**Keywords:** population pharmacokinetic, rivaroxaban, atrial fibrillation, direct oral anticoagulants, Thai patient

## Abstract

Low-dose rivaroxaban has been used in Asian patients with direct oral anticoagulants (DOACs) eligible for atrial fibrillation (AF). However, there are few pharmacokinetic (PK) data in Thai patients to support precise dosing. This study aimed to develop a population PK model and determine the optimal rivaroxaban doses in Thai patients. A total of 240 Anti-Xa levels of rivaroxaban from 60 Thai patients were analyzed. A population PK model was established using the nonlinear mixed-effect modeling approach. Monte Carlo simulations were used to predict drug exposures at a steady state for various dosages. Proportions of patients having rivaroxaban exposure within typical exposure ranges were determined. A one-compartment model with first-order absorption best described the data. Creatinine clearance (CrCl) and body weight significantly affected CL/F and V/F, respectively. Regardless of body weight, a higher proportion of patients with CrCl < 50 mL/min receiving the 10-mg once-daily dose had rivaroxaban exposures within the typical exposure ranges. In contrast, a higher proportion of patients with CrCl ≥ 50 mL/min receiving the 15-mg once-daily dose had rivaroxaban exposures within the typical exposure ranges. The study’s findings suggested that low-dose rivaroxaban would be better suited for Thai patients and suggested adjusting the medication’s dose in accordance with renal function.

## 1. Introduction

Rivaroxaban, one of the direct Factor Xa (FXa) inhibitors, is an oral anticoagulant approved for stroke prevention in patients with direct oral anticoagulants (DOACs) eligible for atrial fibrillation (AF) [1]. The approved doses for AF are 20 and 15 mg administered once daily in patients with normal renal function and moderately impaired renal function, respectively [1]. Rivaroxaban was proved to be non-inferior to warfarin in clinical studies in Caucasians and Asians (the ROCKET AF and the Japanese ROCKET AF studies) [2,3]. High interindividual variability and the unexpected risk of bleeding caused by rivaroxaban are concerns in the Asian population [4]. Although low-dose rivaroxaban has been used based on patient-specific factors in a real-world setting among Asian patients [3,4,5], the precise low dose has never been established in this population. This increases the uncertainty around the clinical practice of rivaroxaban prescription. Additionally, because of the substantial interindividual variability of this medication, selecting the best dose in accordance with patient characteristics may help to increase its effectiveness and safety. However, there is a lack of important pharmacokinetic (PK) data for choosing the right dose. As a result, it can be difficult to precisely dose rivaroxaban for Asian individuals.

Anti-Xa activity is linearly correlated with the plasma concentration of direct FXa inhibitors, including rivaroxaban [6,7], making it useful for predicting clinical outcomes. Studies have shown that individuals with AF who had a higher Anti-Xa levels of rivaroxaban measured at peak concentration reported more bleeding complications following treatment [8,9,10]. In addition, a previous study showed an elevated risk of thrombotic events in patients with low Anti-Xa level measured at trough concentration [11,12]. Therefore, a precise dosage of rivaroxaban would be required to provide optimum exposure. Furthermore, substantial interpatient variability in the trough concentration of rivaroxaban has been observed, which raises the risk of unexpected bleeding [13,14]. Despite the fact that therapeutic drug monitoring is not required for all patients treated with rivaroxaban in a routine clinical care setting [15], it may be useful for patients in certain clinical situations (i.e., major bleeding, surgery, renal failure, thromboembolism, and drug-drug interactions), as well as for a population at a high risk of bleeding, such as the elderly, extremely underweight, and those with renal impairment [13,16].

Previous studies showed high interindividual variabilities of rivaroxaban concentrations [17,18]. Numerous variables, including age, gender, body weight, and renal function, as well as drug-drug interactions (e.g., inhibitors or inducers of CYP3A4 and/or P-glycoprotein), may contribute to the considerable variability in rivaroxaban’s PK [19,20]. Thus, dosage modification of rivaroxaban based on patient characteristics may result in optimum drug exposure and safer dose administration.

The disparities in rivaroxaban dosage needs between ethnic groups have been reported [2,3], which may be attributable to the difference in body weight and renal function [21,22]. The clinical trials in Caucasians (the global ROCKET AF) and Japanese (the Japanese ROCKET AF) demonstrated comparable clinical outcomes for stroke prevention and bleeding events [2,3], despite the fact that the Japanese patients received a lower dose of rivaroxaban (15 mg once daily for patients with normal renal function and 10 mg once daily for patients with moderately impaired renal function) [3]. Population PK studies revealed equivalent simulated area under the plasma concentration-time curve from 0–24 h (AUC_0–24_) and peak plasma concentration (C_MAX_) values for rivaroxaban at 15 mg once day in Japanese and 20 mg once daily in non-Japanese individuals [23,24]. Japanese patients taking 20 mg once daily, on the other hand, showed a larger simulated AUC_0–24_ and C_MAX_ than Caucasian patients receiving the same dosage. These findings supported the use of low-dose rivaroxaban in Japanese patients with DOAC-eligible AF.

With Thai individuals sharing comparable features to Japanese patients, a lower dose of rivaroxaban may be more appropriate. A cohort study in Thai DOAC-eligible AF patients discovered that using the lower Japan-specific dose resulted in a larger proportion of patients meeting the typical exposure ranges of Anti-Xa at peak concentrations, whereas the approved dose was likely to exceed the range [25]. As a result, a lower dose of rivaroxaban is likely to be used in Thai patients. However, the rivaroxaban PK profiles in the Thai population have not been studied. The information on rivaroxaban PK derived from real-world data is required to facilitate dose optimization. Additionally, dosing based on patient characteristics could be utilized to guide the individualized rivaroxaban dose. Thus, this study aimed to develop a population PK model and evaluate factors influencing rivaroxaban PK. The developed model was then applied to design the optimal dose using Monte Carlo simulations.

## 2. Materials and Methods

### 2.1. Study Design

The analysis included data from a previous study comparing Anti-Xa levels at peak and trough between the standard dose and a Japan-specific dose of rivaroxaban in Thai patients [25]. Sixty adult patients with non-valvular AF were included in the previous study at a tertiary hospital that serves as an academic and referral center for northern Thailand between June 2018 and January 2019. Patients with significant renal impairment (CrCl < 15 mL/min) or poor medication adherence were excluded [25].

Rivaroxaban was given to the patients based on their renal function, determined as CrCl. Two dosing strategies, including the standard doses and the Japan-specific doses, were evaluated. The standard doses were 20 mg once daily for CrCl ≥ 50 mL/min and 15 mg once daily for CrCl 15–49 mL/min, while the Japan-specific doses were 15 mg once daily for CrCl ≥ 50 mL/min and 10 mg once daily for CrCl 15–49 mL/min.

All patients were started with the standard dose of rivaroxaban for at least one week. Then the Anti-Xa were collected at peak (2–4 h after taking the dose) and trough concentrations (22–24 h after the last dose). They were then switched to the Japan-specific dose, and the Anti-Xa were measured at peak and trough concentrations after at least one week of rivaroxaban administration.

Ethical approval was given by the Institutional Review Board of the Faculty of Medicine, Chiang Mai University, Chiang Mai, Thailand, and the Institutional Review Board Committee on human research at the University of Phayao, Phayao, Thailand.

### 2.2. Rivaroxaban Quantification

Blood samples were collected into 3.2% sodium citrate tubes and mixed by inverting them for 8–10 min to avoid clotting. The samples were centrifuged at 3000 rpm for 10 min. Plasma was separated and stored at −70 °C until the analysis.

The Anti-Xa activity of rivaroxaban in plasma was measured with the chromogenic method using BIOPHEN^TM^ Heparin LRT kits (Dasit, Milan, Italy) and analyzed on an Automated Blood Coagulation Analyzer (Sysmex CS 2500 System, Siemens Health Care, Milan, Italy). All reagents and instruments were used in accordance with the manufacturers’ instructions. The Anti-Xa was specifically calibrated and translated into values of rivaroxaban concentration [25]. The BIOPHEN^TM^ Rivaroxaban Calibrator kits were used to set the curve for the low range and standard range. These calibrators were used to establish the calibration curves for the Anti-Xa chromogenic assays of rivaroxaban in plasma. The BIOPHEN^TM^ Rivaroxaban Control kits were used to control the curve for the low range and standard range. These controls were used for the quality control of Anti-Xa chromogenic assays of rivaroxaban in plasma.

### 2.3. Population Pharmacokinetic Analysis

The population PK analysis of rivaroxaban was performed using nonlinear mixed effects modelling in NONMEM^®^ software, version 7.4 (Icon Development Solution, Ellicott City, MD, USA). The first-order conditional estimation method with interaction (FOCE-I) was used throughout the model development. Model diagnostics and automated functions for model development were performed using Xpose version 4.0 [26], Pirana [27], and Perl-speaks-NONMEM (PsN; version 5.2.6) [28].

The data availability during an absorption phase was limited since the Anti-Xa activity of rivaroxaban was only obtained at steady-state peak and trough concentration on two different dosing occasions. Thus, a frequentist prior approach ($PRIOR) was implemented to stabilize model parameter estimates [29]. The typical value and inter-individual variability (IIV) of the oral absorption rate constant (ka) from a previous study [23] were applied using a frequentist prior methodology.

The PK parameters were assumed to be log-normally distributed, and their IIVs were introduced as an exponential function. The residual unexplained variability (RUV) was modeled using an additive function.

The difference in objective function value (ΔOFV) was used as a criterion for model discrimination. The ΔOFV of >3.84 and >6.63 were considered statistically significant at a *p*-Value of <0.05 and <0.01, respectively, with one degree of freedom.

The influence of patient characteristics on PK parameters was investigated using a stepwise covariate approach. During stepwise forward inclusion, the ΔOFV of >3.84 was used to include the covariate in the model, and the more stringent criterion (ΔOFV of >6.63) was used in the stepwise backward deletion to retain the covariate in the final model. The following covariates were evaluated: age (years), body weight (kg), body mass index (kg/m^2^), creatinine level (mg/dL), creatinine clearance (CrCl; mL/min, calculated according to Cockcroft and Gault equation) and sex.

Model misspecification and systematic errors were assessed using the goodness-of-fit plots. The ability to detect model misspecifications in goodness-of-fit diagnostics was evaluated using eta and epsilon shrinkages [30]. The robustness of the parameter estimated from the final model was evaluated using nonparametric bootstrap (*n* = 1000). The visual predictive checks (VPC) were performed using 1000 simulations. The 50th percentile of the observed data was overlaid with the simulated 90% prediction interval to evaluate the model’s predictive performance.

### 2.4. Simulations

Monte Carlo simulations were performed using the final model to evaluate three different rivaroxaban doses, i.e., 10, 15, and 20 mg, once daily. Rivaroxaban exposures at a steady state (i.e., peak concentration; C_MAX_, trough concentration; C_MIN_ and 24 h area under the concentration-time curve; AUC_0–24_) were calculated. The 5th and 95th percentile ranges of C_MAX_ (184–343 ng/mL), C_MIN_ (12–137 ng/mL), and AUC_0–24_ (1860–5434 ng·h/mL) reported in DOAC-eligible AF patients were used as the typical exposure ranges [19]. For each subgroup of patients based on CrCl and body weight (i.e., 15–29, 30–49, 50–69, 70–89, and 90–110 mL/min for CrCl and 15–29, 30–59, 60–89, and 90–119 kg for body weight), 1000 in silico patients were simulated. The percentage of patients with C_MAX_, C_MIN_, and AUC_0–24_ values that were within the typical exposure ranges were calculated.

## 3. Results

### 3.1. Population Pharmacokinetic Analysis

A total of 240 Anti-Xa levels measured at peak and trough concentrations of rivaroxaban were obtained from 60 Thai DOAC-eligible AF patients. The summary of patient characteristics is presented in Table 1.

The PK of rivaroxaban was described by a one-compartment model with first-order absorption and first-order elimination. The estimation of CL/F was 4.19 L/h with IIV of 21.9%, V/F was 37.5 L, and ka was 0.697 h-1 with IIV of 75.9%. Parameter estimates from the final model are presented in Table 2.

CrCl was a significant covariate for rivaroxaban CL/F (ΔOFV = −9.96, *p*-Value < 0.01). A power coefficient of 0.278 was used to define their association, showing that patients with higher CrCl have higher rivaroxaban CL/F. Additionally, body weight was a significant covariate for V/F (ΔOFV = −7.33, *p*-Value < 0.01) with a power coefficient of 0.412, indicating that patients with higher body weight have higher rivaroxaban V/F.

The goodness of fit plots showed that the final model adequately describes the data (Figure 1). The simulation-based diagnostic of the final model showed an adequate predictive performance (Figure 2). Eta shrinkage for CL/F (5.9%) and epsilon shrinkage (17.1%) were <20%, indicating robust individual parameter estimates [30].

### 3.2. Simulations

Figure 3 and Figure 4 show the simulated C_MAX_ and C_MIN_ for different dosage regimens, stratified by CrCl and body weight. As body weight did not effect on the simulated AUC_0–24_, the simulated AUC_0–24_ by CrCl was displayed (Figure 5). The percentage of simulated patients with rivaroxaban exposures within the typical exposure ranges under various scenarios is summarized in Figure 6. The C_MAX_, C_MIN_, and AUC_0–24_ values of patients who received 20 mg of rivaroxaban once daily were greater than the typical exposure ranges, particularly in those with CrCl less than 50 mL/min.

In comparison to the other dosing groups, the dosage of 10 mg once daily had the highest proportion of patients with C_MAX_, C_MIN_, and AUC_0–24_ values within the typical exposure ranges for patients with CrCl between 15–29 and 30–49 mL/min, regardless of body weight. The proportion of patients with C_MAX_, C_MIN_, and AUC_0–24_ within the typical exposure ranges was greater for patients with CrCl 50–69, 70–89, and 90–110 mL/min who took 15 mg once a day, regardless of body weight.

## 4. Discussion

Because of differences in patient characteristics, the Japanese patients had greater rivaroxaban concentrations when treated with the approved dosage [23,24]. The renal function of Asian patients with AF, as measured by CrCl, was lower than that of Caucasians [31,32]. As a result, it is likely that a lower dose of rivaroxaban gives enough exposure for Asians [23,24,33,34], reducing the risk of bleeding [5,35]. However, the PK data needed to optimize doses in Thai and Asian populations is scarce. Additionally, the high cost of rivaroxaban restricts access to therapy in resource-limited settings, including Thailand. Due to concerns about the risk of bleeding and cost, a lower dosage of DOACs has been prescribed for Thai patients [36]. Thus, the optimal low dose of rivaroxaban provided by this study could make rivaroxaban more accessible and safer for Thai patients. The current study developed a population PK model of rivaroxaban in Thai DOAC-eligible AF patients. To find the optimal dose of rivaroxaban, simulations were performed to predict drug exposure depending on significant covariates.

The rivaroxaban PK in Thai DOAC-eligible AF patients was adequately described by a one-compartment model with first-order absorption and elimination, consistent with previous studies in AF patients [20,23,24,33,34,37]. The estimation of CL/F (4.19 L/h) in Thai DOAC-eligible AF patients was slightly lower than that reported in other ethnicities, including Caucasians (5.58–6.10 L/h) [20,37]. When compared to other Asian populations, the CL/F in this study is comparable to Japanese (4.72–4.73 L/h) [23,24] but lower than Chinese (5.03–7.39 L/h) [33,34,38]. The estimated V/F (37.5 L) was lower than that reported in other studies (40.3–79.7 L) [23,24,33,34,37]. The discrepancies in PK values between ethnic groups might be attributable to variability in patient characteristics. Due to the small number of samples collected during the distribution phase, we were unable to determine the IIV of the V/F. The parameterized ka using a frequentist prior approach was 0.697 h^−1^, which is closed to the value (0.600–0.617 h^−1^) reported in Japanese [23,24].

In the covariate analysis, the final model revealed that CL/F and V/F of rivaroxaban increase by increasing CrCl and body weight, respectively. Several previous studies showed the significant effect of CrCl on rivaroxaban CL/F [20,24,34,39,40,41,42], thus, our finding confirmed this relationship. Because rivaroxaban is partly excreted through the kidneys, reduced renal function results in lower drug clearance [19]. The Cockcroft–Gault equation, which is preferably used to determine the renal function and is expressed as CrCl, is indicative of rivaroxaban dose adjustment [1,19].

Age, gender, or body weight had no influence on rivaroxaban CL/F. These results are consistent with earlier findings showing that the C_MAX_ of rivaroxaban was unaltered in patients with high body mass but increased by 24% in those with severely low body mass (50 kg), resulting in a prolonged prothrombin time [43]. A prior study indicated that elderly patients were exposed to 41 percent more rivaroxaban than young subjects [44]. However, the increasing exposure with age was mostly attributable to a lower clearance as a result of decreased renal function [37,44]. Moreover, these factors are employed in the Cockcroft-Gault equation to determine CrCl.

It has been determined that body size (i.e., lean body mass and body surface area) influences the PK characteristics of rivaroxaban [20,34,37,39]. One research involving healthy volunteers demonstrated that BMI influences the V/F of rivaroxaban [38]. Moreover, the results from previous studies showing that CrCl was the significant covariate influencing rivaroxaban CL/F, whereas other body size measurements alone were not found to significantly impact rivaroxaban CL/F [40,41]. In our study, BMI was not detected as a significant covariate for either V/F or CL/F (*p* > 0.05) during covariate analysis. However, body weight was a significant covariate for V/F (*p* < 0.01) with an improvement in the goodness of fit. In clinical practice, body weight is more readily available than other weight measurements. Thus, using body weight as a measurement of body size would be more practical.

In our analysis, the IIV of V/F could not be estimated from the data; thus, the V/F was estimated without its IIV. Although the impact of body weight in explaining part of the IIV of V/F cannot be determined, weight was kept in the final model as a significant covariate on V/F as it showed a significantly decreased OFV and improved the overall model fit.

Rivaroxaban exposure has been associated with clinical outcomes and may aid in predicting the benefit–risk profile [13,16]. Peak Anti-Xa concentrations have been linked to bleeding complications [8,9,10], whereas trough concentrations have been related to the occurrence of thromboembolic events [11,12].

Based on the results from simulations, it was shown that the use of the approved dose in the Thai population resulted in higher rivaroxaban exposures. For patients with normal renal function (CrCl ≥ 50 mL/min) irrespective of body weight, a reduced dose of 15 mg resulted in a high proportion of patients having rivaroxaban exposure within the typical exposure ranges. This reduced dose is considered more appropriate for Thai DOAC-eligible AF patients with normal renal function, which supports the results from population PK studies in Asian patients, including Japanese [23,24] and Chinese [33,34].

Patients with renal impairment had a decrease in the CL/F, leading to increased rivaroxaban exposure [19]. Based on the simulation results, the C_MAX_, C_MIN_, and AUC_0–24_ were higher in patients with lower CrCl, which are in line with previous findings [20,37]. Our simulation results showed that the predicted C_MAX_ was decreased with increasing body weight in the same CrCl range group. In contrast, individuals with higher body weight are more likely to have a higher predicted C_MIN_ within the same group of CrCl patients, which may be owing to an increase in V/F as body weight increases, resulting in a decrease in the elimination rate constant (ke). Similar trends in C_MAX_ and C_MIN_ alterations were seen in Chinese patients in a prior investigation [34].

A previous study, which looked at a wider range of CrCl (i.e., 15–29, 30–69, 70–159, and 160–250 mL/min) than the phase III clinical trial, showed that for individuals with CrCl of 15–29 mL/min, the dosage should be lowered from 15 to 10 mg [45]. Although rivaroxaban dosing has not been proven for patients with CrCl 15–29 mL/min [1], a dose reduction may be warranted for people in this group who require this medication. The results from our study suggested a daily dose of 10 mg for Thai DOAC-eligible AF patients with CrCl 15–29 mL/min, regardless of body weight. This recommendation is consistent with the dosage indicated for this population in previous research [45].

The previous integrated PK/PD study of rivaroxaban in Chinese patients suggested that the median peak Anti-Xa level at a dose of 10 mg was within the expected range for patients with CrCL 30–49 mL/min but not at a dose of 15 mg [38]. As a result, dosage modification based on body weight is unnecessary [38]. Results from our simulations confirmed a lower rivaroxaban dose of 10 mg should be appropriate for patients with poor renal function (CrCl of 30–49 mL/min) regardless of body weight. In summary, for patients with renal impairment, CrCL < 50 mL/min, a daily dose of 10 mg of rivaroxaban is recommended.

In this investigation, rivaroxaban concentrations were determined indirectly using commercially available Anti-Xa assays (BIOPHEN^TM^ Heparin LRT) with validated specific rivaroxaban calibrators. Despite the fact that liquid chromatography with tandem mass spectrometry (LC-MS/MS) is the gold standard approach for direct detection of rivaroxaban plasma concentrations [46], it is time-consuming, technically demanding, and not always available. The technique comparison research revealed a correlation between the BIOPHEN^TM^ Heparin LRT and LC-MS/MS test at all rivaroxaban concentrations (r^2^ = 0.97) [47]. Thus, Anti-Xa tests can be used to detect rivaroxaban levels indirectly in clinical practice.

This study has some limitations. First, data during the absorption phase were limited. Thus, the prior method was used to estimate ka and its variability, which indicated the reliability of parameter estimates. Second, the number of patients concomitantly receiving inducers and inhibitors of CYP3A4 and/or P-glycoprotein was small, so the effect of the drug-drug interactions [19,20,48] could not be evaluated. Third, we did not explore other covariates (i.e., genetic polymorphisms, hepatic impairment) which can cause PK variability of rivaroxaban [34,42,49,50]. A previous study revealed that mild hepatic impairment decreased drug clearance and increased drug exposure, resulting in a prolonged prothrombin time [50]. Patients with moderate or severe hepatic impairment (Child-Pugh B and C) should thus avoid using rivaroxaban. Although a prior study determined that total bilirubin was a significant covariate for baseline prothrombin time, its effect on the PK of rivaroxaban was not identified [33,34]. Due to a lack of data, the effect of hepatic function was not explored in our study. Considering the aforementioned factors, further research should be undertaken.

Fourth, the typical exposure ranges of rivaroxaban were established from prior research on patients with non-valvular AF [19]. As the therapeutic window associated with the risk-benefit profile of rivaroxaban is unknown, the optimal dosage recommended in this study is the dose that provides equivalent rivaroxaban exposure to the typical exposure ranges, which may not be the clinically optimal dose. Finally, the impact of a reduced dose on clinical outcomes was not investigated in this study. Several previous studies found that low-dose rivaroxaban, compared to warfarin, was associated with a lower risk of stroke or systemic embolism and major bleeding [5,35,51,52]. However, there is evidence of a higher risk of ischemic stroke without a lower risk of bleeding when a lower dose of rivaroxaban was given to Asian patients [53]. Further investigations are needed to determine the clinical relevance of low-dose rivaroxaban regarding effectiveness for stroke prevention and safety.

## 5. Conclusions

In conclusion, a population PK model of rivaroxaban PK in Thai DOAC-eligible AF patients confirmed the effect of renal function on CL/F and the effect of body weight on V/F. Simulations suggested that low-dose rivaroxaban may be more appropriate for Thai patients, and dosages of rivaroxaban depending on renal function were proposed for Thai patients. A daily 10 mg dose was proposed for Thai DOAC-eligible AF patients with CrCl < 50 mL/min, regardless of body weight. For Thai DOAC-eligible AF patients with CrCl ≥ 50 mL/min, a daily dose of 15 mg was recommended.

## Figures and Tables

**Figure 1 pharmaceutics-14-01744-f001:**
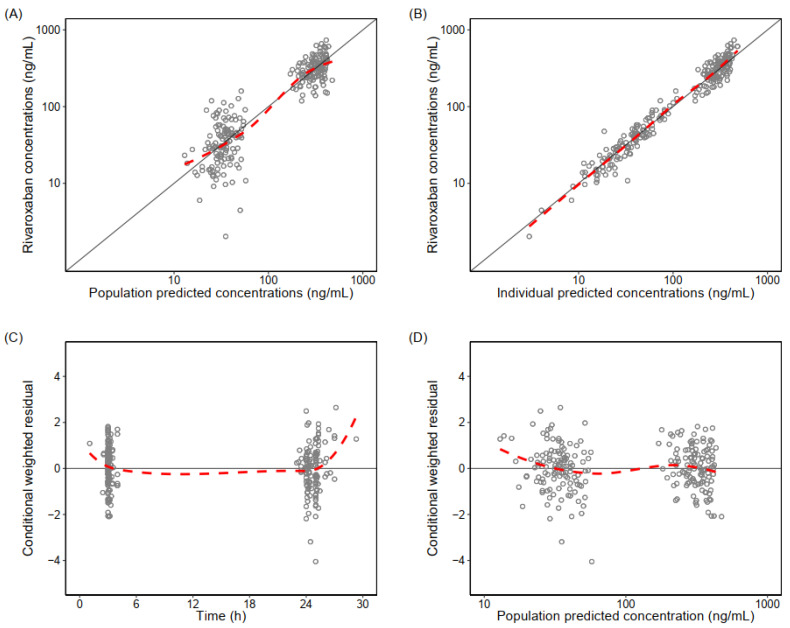
Goodness-of-fits of the final population pharmacokinetic model of rivaroxaban. (**A**) Observed rivaroxaban concentrations vs. population predictions, (**B**) observed rivaroxaban concentrations vs. individually predicted concentrations, (**C**) conditionally weighted residual vs. time, and (**D**) conditionally weighted residual vs. population predictions. The open circles represent the observed rivaroxaban concentrations. The solid black lines are the line of identity or zero-line. The dashed red lines are loess smooth lines (trend lines).

**Figure 2 pharmaceutics-14-01744-f002:**
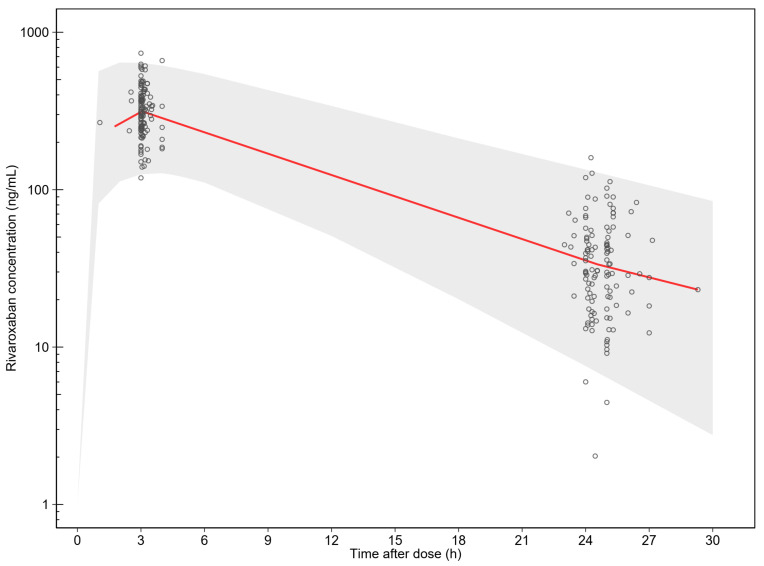
The simulated 90% prediction interval from the final population pharmacokinetic model of rivaroxaban. The open circles represent the observed rivaroxaban concentrations. Solid red line represents the 50th percentile of the observations. The shaded area represents the 90% prediction interval of the simulations (*n* = 1000).

**Figure 3 pharmaceutics-14-01744-f003:**
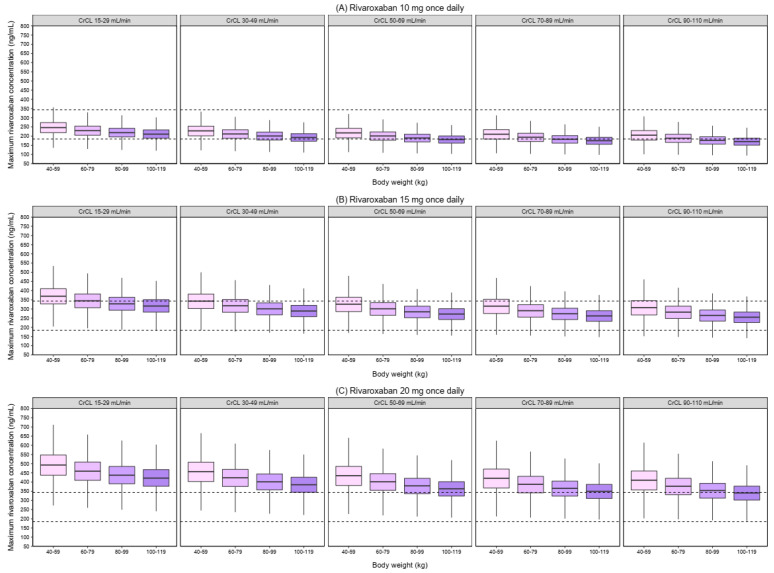
Simulated maximum concentration (C_MAX_) of rivaroxaban with different dosing regimens. The boxplots represent the predicted C_MAX_ stratified by creatinine clearance and body weight. (**A**) rivaroxaban 10 mg once daily. (**B**) rivaroxaban 15 mg once daily. (**C**) rivaroxaban 20 mg once daily. The dashed lines represent the 5th and 95th percentile ranges of the typical C_MAX_ (184–343 ng/mL) reported in patients with non-valvular atrial fibrillation receiving 20 mg of rivaroxaban [19].

**Figure 4 pharmaceutics-14-01744-f004:**
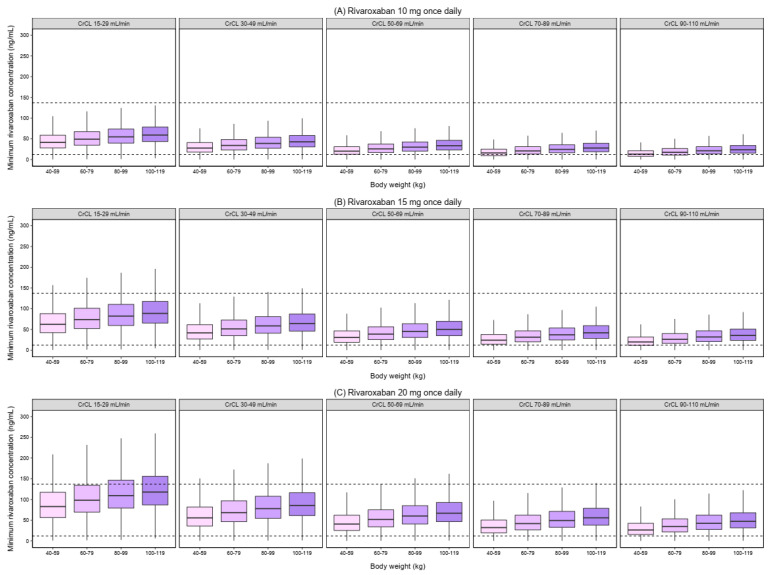
Simulated minimum concentration (C_MIN_) of rivaroxaban with different dosing regimens. The boxplots represent the predicted C_MIN_ stratified by creatinine clearance and body weight. (**A**) rivaroxaban 10 mg once daily. (**B**) rivaroxaban 15 mg once daily. (**C**) rivaroxaban 20 mg once daily. The dashed lines represent the 5th and 95th percentile ranges of the typical C_MIN_ (12–137 ng/mL) reported in patients with non-valvular atrial fibrillation receiving 20 mg of rivaroxaban [19].

**Figure 5 pharmaceutics-14-01744-f005:**
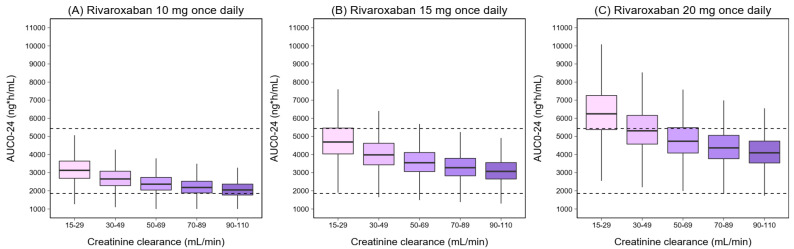
Simulated AUC_0–24_ of rivaroxaban with different dosing regimens. The solid black lines represent the median AUC_0–24_ The shaded areas represent the 90% prediction interval of the model. The dashed lines represent the 5th and 95th percentile ranges of the typical AUC_0–24_ (1860–5434 ng·h/mL) reported in patients with non-valvular atrial fibrillation receiving 20 mg of rivaroxaban [19].

**Figure 6 pharmaceutics-14-01744-f006:**
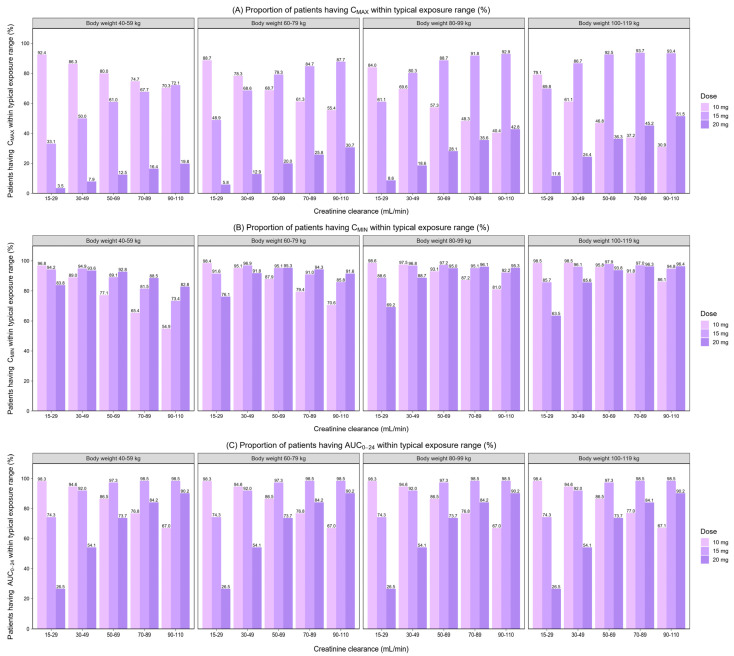
Proportion of patients having predicted C_MAX_, C_MIN_, and AUC_0–24_ within the typical exposure ranges reported in patients with non-valvular atrial fibrillation [19] (1000 simulations per subgroup).

**Table 1 pharmaceutics-14-01744-t001:** Summary of patient characteristics (*n* = 60).

Characteristics	Value
Male, *n* (%)	38 (63.3)
Age (y), mean (SD)	69.4 (9.2)
Body weight (kg), mean (SD)	64.0 (14.1)
Body mass index (kg/m^2^), median (Q_1_, Q_3_)	24.2 (21.5, 26.9)
Creatinine (mg/dL), mean (SD)	1.1 (0.3)
Creatinine clearance (mL/min), mean (SD)	59.0 (22.8)
CHADS_2_ score, median (Q_1_, Q_3_)	2 (1, 2)
CHA_2_ DS_2_-VASC score, median (Q_1_, Q_3_)	3 (2, 4)
HAS-BLED score, median (Q_1_, Q_3_)	2 (1, 2)
Underlying disease, *n* (%)	
Hypertension	46 (76.7)
Dyslipidemia	35 (58.3)
Diabetes	16 (26.7)
Congestive heart failure	15 (25.0)
Ischemic heart disease	8 (13.3)
Ischemic stroke	7 (11.7)
Concomitant medications, *n* (%)	
Dronedarone	3 (5.0)
Amiodarone	1 (1.7)
Aspirin	1 (1.7)
Clopidogrel	1 (1.7)

**Table 2 pharmaceutics-14-01744-t002:** Parameter estimates of rivaroxaban in Thai patients with non-valvular atrial fibrillation obtained from the final model and bootstrap analysis.

Parameters	Final Model (NONMEM)		Bootstrap Analysis (*n* = 1000 Samples)	
	Estimates ^a^ [RSE%]	95% CI *	Median ^b^ [RSE%]	95% CI **
CL/F (L/h)	4.19 [3.8%]	3.88–4.50	4.21 [4.08]	3.88–4.55
V/F (L)	37.5 [4.7%]	34.03–40.97	37.59 [4.62%]	34.0–40.8
ka (h^−1^)	0.697 [10.7%]	0.550–0.844	0.699 [5.06%]	0.623–0.764
CrCl on CL/F ^c^	0.277 [29%]	0.120–0.434	0.277 [35.2%]	0.054–0.480
WT on V/F ^d^	0.412 [35.7%]	0.124–0.70	0.413 [28.1%]	0.149–0.639
IIV of CL/F (%CV)	21.94 [21.3%]	16.67–26.24	21.24 [14.7%]	14.8–27.5
IIV of ka (%CV)	75.91 [10.1%]	66.39–85.10	75.81 [1.08%]	74.1–78.1
RUV, additive (mg/L)	0.092 [11.7%]	0.071–0.114	0.0926 [9.44%]	0.064–0.131

Abbreviations; CL/F, apparent oral clearance; V/F, apparent volume of distribution; ka, absorption rate constant; IIV, interindividual variability; RUV, residual unexplained variability; %CV, percent coefficient of variation; CI, confidence interval; SE, standard error; CrCl, creatinine clearance (mL/min); WT, body weight (kg). ^a^ Population mean values was estimated by NONMEM. * 95%CI = estimated value ± (1.96 × SE). ^b^ Median values was calculated from the non-parametric bootstrap results (*n* = 1000). ** 95%CI = 2.5th and 97.5th percentiles of the bootstrap parameter estimates. ^c^ Calculated as; CL/F = 4.19 × (CrCl/57.5)^0.277^. ^d^ Calculated as; V/F = 37.5 × (WT/63)^0.412^.

## Data Availability

The data that support the findings of this study are available from the corresponding author upon request.

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
