# Peer review of "Population Pharmacokinetics and Dose Optimization Based on Renal Function of Rivaroxaban in Thai Patients with Non-Valvular Atrial Fibrillation"

_pharmaceutics, 2022, doi:10.3390/pharmaceutics14081744_

Round 1

Reviewer 1 Report

The manuscript introduces the population pharmacokinetics and dose optimization of rivaroxaban in Thai patients based on the creatinine clearance. There are shortcomings in the experimental design of this study, and the content is thin. So I believe the manuscript is not appropriate for Pharmaceutics, at least for now.

1.      Regarding the dosage of rivaroxaban, there are guidance documents, such as https://reference.medscape.com/drug/xarelto-rivaroxaban-999670, of which, the creatinine clearance for the dose guidance also has definite rules. So, what is the main innovation of this paper? Is the population pharmacokinetic evaluation and dose optimization of rivaroxaban for patients with atrial fibrillation in Thailand, just the distribution of people?

2.      Despite the three limitations mentioned at the end of the article (Page 11, Line 325-330), it is well-known that rivaroxaban mainly experiences CYP3A4/2J2 metabolism and p-GP /BCRP transport in vivo, and the hepatic clearance accounts for about 65% and kidney excretion accounts for about 35% (PMID: 30581412). Compared to renal function, liver function cannot be ignored in the pharmacokinetic behavior variations of rivaroxaban. Therefore, if the liver function data of the 60 patients with atrial fibrillation involved in this study are available, it is strongly suggested to include the liver functions, such as Moderate hepatic impairment (child-pugh B) and Severe hepatic impairment (child-pugh C), in the relevant analysis.

3.      Since this paper only explored the effect of creatinine clearance on the pharmacokinetic behavior of rivaroxaban in the Thai population, it is strongly recommended that keywords in renal function or creatinine clearance be included in the title.

4.      For the population pharmacokinetic study, the number of 60 patients is relatively small, and it is suggested to increase to more than 200 patients.

Author Response

The manuscript introduces the population pharmacokinetics and dose optimization of rivaroxaban in Thai patients based on the creatinine clearance. There are shortcomings in the experimental design of this study, and the content is thin. So I believe the manuscript is not appropriate for Pharmaceutics, at least for now.

We thank the reviewer for valuable comments. We have revised the paper based on the reviewers' recommendations in order to make it suitable for Pharmaceutics.

  1. Regarding the dosage of rivaroxaban, thereare guidance documents, such as https://reference.medscape.com/drug/xarelto-rivaroxaban-999670, of which, the creatinine clearance for the dose guidance also has definite rules. So, what is the main innovation of this paper? Is the population pharmacokinetic evaluation and dose optimization of rivaroxaban for patients with atrial fibrillation in Thailand, just the distribution of people?

We appreciate your feedback. We concur that the dose recommendations for rivaroxaban are driven by current recommendations based on worldwide clinical studies. In addition to current clinical evidence, pharmacokinetic profiles of rivaroxaban derived from real-world data and adjusted for patient characteristics are necessary to guarantee adequate drug exposures and safer.

Asian patients were shown to have higher rivaroxaban concentration when the approved dose was administered. Consequently, it is feasible that a lower dose of rivaroxaban might offer enough exposure to rivaroxaban in the Asian population, allowing for less bleeding issues. Although the lower dose of rivaroxaban is routinely prescribed to Asian patients, the precise low-dose has never been established in this population. Thus, determining the exact optimal low-dose of rivaroxaban for Asia patients including Thai is important.

We have revised “Introduction” to address the important of this research regarding rivaroxaban treatment in Asian population including Thai. (line 44-55)

“Rivaroxaban was proved to be non-inferior to warfarin in clinical studies in Caucasians and Asians (the ROCKET AF and the Japanese ROCKET AF studies) [2, 3]. High inter-individual variability and the unexpected risk of bleeding caused by rivaroxaban are concerned in Asian population [4]. Although, the low-dose rivaroxaban has been used based on patient-specific factors in a real-world setting among Asian patients [3-5], the precise low-dose has never been established in this population. This increases the uncertainty around the clinical practice of rivaroxaban prescription. Additionally, because of the substantial interindividual variability of this medication, selecting the best dose in accordance with patient characteristics may help to increase its effectiveness and safety. However, there is a lack of important pharmacokinetic data for choosing the right dose. As a result, it can be difficult to precisely dose rivaroxaban for Asian individuals.”

We have also revised “Discussion” to provide the clinically relevant for Thai patients. (line 314-319)

“Additionally, the high cost of rivaroxaban restricts access to therapy in resource-limited settings, including Thailand. Due to concerns about the risk of bleeding and cost, a lower dosage of DOACs has been prescribed for Thai patients [36]. Thus, the optimal low dose of rivaroxaban provided by this study could make rivaroxaban more accessible and safer for Thai patients.”

  1. Despite the three limitations mentioned at the end of the article (Page 11, Line 325-330), it is well-known that rivaroxaban mainly experiences CYP3A4/2J2 metabolism and p-GP /BCRP transport in vivo, and the hepatic clearance accounts for about 65% and kidney excretion accounts for about 35% (PMID: 30581412). Compared to renal function, liver function cannot be ignored in the pharmacokinetic behavior variations of rivaroxaban. Therefore, if the liver function data of the 60 patients with atrial fibrillation involved in this study are available, it is strongly suggested to include the liver functions, such as Moderate hepatic impairment (child-pugh B) and Severe hepatic impairment (child-pugh C), in the relevant analysis.

Thank you for your suggestions and comments. We agree that liver function and stages of hepatic impairment may contribute to the pharmacokinetic variability of rivaroxaban. However, liver function was not assessed in our investigation. Further research is required to determine the influence of liver function and stages of hepatic impairment on the pharmacokinetic variability of rivaroxaban. Consequently, we have addressed this as one of our limits under Discussion. (lines 402-410)

“A previous study revealed that mild hepatic impairment (Child-Pugh B) decreased drug clearance and increased drug exposure, resulting in a prolonged prothrombin time [48]. Patients with moderate or severe hepatic impairment (Child-Pugh B and C) should thus avoid using rivaroxaban. Although a prior study determined that total bilirubin was a significant covariate for baseline prothrombin time, its effect on the pharmacokinetics of rivaroxaban was not identified [33, 34]. Due to a lack of data, the effect of hepatic function was not explored in our study. Considering the aforementioned factors, further research should be undertaken.”

  1. Since this paper only explored the effect of creatinine clearance on the pharmacokinetic behavior of rivaroxaban in the Thai population, it is strongly recommended that keywords in renal function or creatinine clearance be included in the title.

Thank you for your suggestions. We have revised the title to include the keyword: renal function:

"Population pharmacokinetics and dose optimization based on renal function of rivaroxaban in Thai patients with non-valvular atrial fibrillation".

  1. For the population pharmacokinetic study, the number of 60 patients is relatively small, and it is suggested to increase to more than 200 patients.

Thank you for your comments and suggestions. We do agree that the number of 60 patients is relatively small and larger sample size could provide more precise parameter estimates. However, a prior study revealed that two blood samples from five individuals can provide a good approximation of pharmacokinetic characteristics (Mahmood I et al.). The required sample size for a population pharmacokinetic research is contingent on several factors including methods of analysis, parameter of choice, and sampling design.  

Prior research by Ogungbenro K et al. demonstrated that for a one-compartment, first-order absorption model, assuming Ka is the parameter of choice, the sample sizes required to estimate the 95% confidence interval within a 20% precision level with a power of 0.9 using the FO, FOCE, and FOCE/INTERACTION methods in NONMEM and WINBUGS for a design involving sampling at 0.01, 7.75, and 12 h are 20, 30, and 30, respectively. In addition, the powers to estimate the 95% confidence interval to within 20% precision were more than 0.95 for all estimation methods when the sample size was greater than 40 (with 3 blood samples). Based on these results, we are confident that the sample size of 60 patients in this study is sufficient for estimating the pharmacokinetic parameters of the population with precision.

Reference:

Mahmood I, Duan J. Population pharmacokinetics with a very small sample size. Drug Metabol Drug Interact. 2009;24(2-4):259-74. doi: 10.1515/dmdi.2009.24.2-4.259. PMID: 20408503.

Ogungbenro K, Aarons L. How many subjects are necessary for population pharmacokinetic experiments? Confidence interval approach. Eur J Clin Pharmacol. 2008;64(7):705-13.

Reviewer 2 Report

 In this study, the authors present a pharmacokinetic model for rivaroxaban concentrations in Thai patients and suggest optimal rivaroxaban doses. This is a single centre study of 60 patients with atrial fibrillation. The paper is well written, but I have some comments and suggestions for improvement:

-        Throughout the paper, the term "reference range" is used incorrectly. The reference range refers to values obtained in patients not receiving rivaroxaban. In the case of rivaroxaban, the reference range is usually < 20 or below. The authors are probably referring to a target range that is not known for rivaroxaban. What we have from the literature are typical exposure ranges. Therefore, the term needs to be replaced throughout the manuscript and also considered when discussing optimal rivaroxaban dosing in Thai patients. This study has shown the doses with which comparable rivaroxaban blood concentrations can be achieved, which may not necessarily be the clinically optimal doses.

-        The statement that "a high anti-Xa level of rivaroxaban measured at peak concentration is related to a higher risk of bleeding complications, while a low anti-Xa level measured at trough concentration is likely to represent an increased risk of thromboembolic events" is too simplistic. Expand and add additional references (for example, DOI: 10.1007/s00228-019-02693-2).

-        The laboratory section should be expanded to include data on blood collection, tube type, plasma processing, and data on rivaroxaban calibrators and controls.

-        Although the study population is described in a cited reference, a brief description should also be provided in this paper. Were they consecutive patients? How were they randomised between the standard and Japan-specific doses? A table of patient characteristics in the Results section would be very informative.

-        Did the 15-mg group (Figures 3 and 4) include both patients whose standard dose was reduced because of a low CrCl level and patients with a Japan-specific dose and normal renal function? This should be explained.

-        Figure 3: please add that the 5th to 95th percentile of rivaroxaban concentrations (dashed lines) was obtained in patients receiving 20 mg rivaroxaban (similar for Figures 4 and 5).

-        Table 2 is difficult to read and therefore not very informative. An attempt should be made to find a graphical representation.

-        The reference list should be changed: Several references are redundant and can be omitted (for example, 35 and 36), some are inappropriate (for example, reference #38 is a review and not an original paper, and reference #39 is inappropriate), and additional references should be included to further explain the role of rivaroxaban concentration in complications. Reference DOI: 10.1177/0091270006296058 should also be included and commented on.

-        The discussion should be shortened and portions that are repetitions of the text in the introduction (e.g., lines 285 - 290) should be removed.

-        Line 246: "Asian patients" are actually Japanese patients. Please be specific.

-        I suggest shortening the title to "Population pharmacokinetics and dose optimization of rivaroxaban in Thai patients with non-valvular atrial fibrillation" as it is sufficiently informative.

-        Anti-Xa is the standard abbreviation for the test.

-        DOAC is preferred over NOAC according to ISTH.

Author Response

Reviewer 2:

In this study, the authors present a pharmacokinetic model for rivaroxaban concentrations in Thai patients and suggest optimal rivaroxaban doses. This is a single centre study of 60 patients with atrial fibrillation. The paper is well written, but I have some comments and suggestions for improvement:

  1. Throughout the paper, the term "reference range" is used incorrectly. The reference range refers to values obtained in patients not receiving rivaroxaban. In the case of rivaroxaban, the reference range is usually < 20 or below. The authors are probably referring to a target range that is not known for rivaroxaban. What we have from the literature are typical exposure ranges. Therefore, the term needs to be replaced throughout the manuscript and also considered when discussing optimal rivaroxaban dosing in Thai patients. This study has shown the doses with which comparable rivaroxaban blood concentrations can be achieved, which may not necessarily be the clinically optimal doses.

Thank you for highlighting these concerns. The word "reference ranges" has been substituted throughout the document with "average exposure ranges." Discussion regarding the the optimal rivaroxaban dosing in Thai patients were included under “Limitations (Discussion)” (lines 411-415).

“Fourth, the typical exposure ranges of rivaroxaban were established from a prior research of patients with non-valvular AF [19]. As the therapeutic window associated with the risk-benefit profile of rivaroxaban is unknown, the optimal dosage recommended in this study is the dose that provides equivalent rivaroxaban exposure to the typical exposure ranges, which may not be the clinically optimal doses.”

  1. The statement that "a high anti-Xa level of rivaroxaban measured at peak concentration is related to a higher risk of bleeding complications, while a low anti-Xa level measured at trough concentration is likely to represent an increased risk of thromboembolic events" is too simplistic. Expand and add additional references (for example, DOI: 10.1007/s00228-019-02693-2).

Thank you for your comment. We have revised “Introduction” and included the additional references as suggested. (line 61-73)

“Studies have shown that individuals with AF who had a higher Anti-Xa level of rivaroxaban measured at peak concentration reported more bleeding complications following treatment [8-10]. In addition, a previous study showed an elevated risk of thrombotic events in patients with low Anti-Xa level measured at trough concentration [11,12]. Therefore, a precise dosage of rivaroxaban would be required to provide optimum exposure. However, substantial interpatient variability in the trough concentration of rivaroxaban has been observed, which raises the risk of unexpected bleeding [13,14]. Despite the fact that therapeutic drug monitoring is not required for all patients treated with rivaroxaban in a routine clinical care setting [15], it may be useful for patients in certain clinical situations (i.e., major bleeding, surgery, renal failure, thromboembolism, and drug-drug interactions), as well as for a population at a high risk of bleeding, such as the elderly, extremely underweight, and those with renal impairment [13, 16].”

References

  1. Jakowenko N, Nguyen S, Ruegger M, Dinh A, Salazar E, Donahue KR. Apixaban and rivaroxaban anti-Xa level utilization and associated bleeding events within an academic health system. Thromb Res. 2020;196:276-82.
  2. Nosáľ V, Petrovičová A, Škorňová I, Bolek T, Dluhá J, Stančiaková L, et al. Plasma levels of direct oral anticoagulants in atrial fibrillation patients at the time of embolic stroke: a pilot prospective multicenter study. Eur J Clin Pharmacol. 2022;78(4):557-64.
  3. Miklič M, Mavri A, Vene N, Söderblom L, Božič-Mijovski M, Pohanka A, et al. Intra- and inter- individual rivaroxaban concentrations and potential bleeding risk in patients with atrial fibrillation. Eur J Clin Pharmacol. 2019;75(8):1069-75.
  4. Sennesael AL, Larock AS, Douxfils J, Elens L, Stillemans G, Wiesen M, et al. Rivaroxaban plasma levels in patients admitted for bleeding events: insights from a prospective study. Thromb J. 2018;16:28.
  5. Cuker A. Laboratory measurement of the non-vitamin K antagonist oral anticoagulants: selecting the optimal assay based on drug, assay availability, and clinical indication. J Thromb Thrombolysis. 2016;41(2):241-7.
  6. Ikeda K, Tachibana H. Clinical implication of monitoring rivaroxaban and apixaban by using anti-factor Xa assay in patients with non-valvular atrial fibrillation. J Arrhythm. 2016;32(1):42-50.

  1. The laboratory section should be expanded to include data on blood collection, tube type, plasma processing, and data on rivaroxaban calibrators and controls.

Thank you for your comments and suggestions. We have included the laboratory method under the “Method: rivaroxaban quantification”. (line 129-142)

“Blood samples were collected into 3.2% sodium citrate tubes and mixed by in-verting them for 8–10 min to avoid clotting. The samples were centrifuged at 3,000 rpm for 10 min. Plasma was separated and stored at -70 0C until the analysis.

The Anti-Xa activity of rivaroxaban in plasma was measured with the chromogenic method using BIOPHENTM Heparin LRT kits (Dasit, Milan, Italy) and analyzed on an Automated Blood Coagulation Analyzer (Sysmex CS 2500 System, Siemens Health Care, Milan, Italy). All reagents and instruments were used in accordance with the manufacturers’ instructions. The Anti-Xa was specifically calibrated and translated into values of rivaroxaban concentration [25]. The BIOPHENTM Rivaroxaban Calibrator kits were used to set the curve for the low range and standard range. These calibrators were used to establish the calibration curves for the Anti-Xa chromogenic assays of rivaroxaban in plasma. The BIOPHENTM Rivaroxaban Control kits were used to control the curve for the low range and standard range. These controls were used for the quality control of An-ti-Xa chromogenic assays of rivaroxaban in plasma.”

  1. Although the study population is described in a cited reference, a brief description should also be provided in this paper. Were they consecutive patients? How were they randomised between the standard and Japan-specific doses? A table of patient characteristics in the Results section would be very informative.

Thank you for your suggestions and comments. The population PK analysis was conducted using data from prior observational research at an urban teaching hospital of tertiary care. Adult patients treated with rivaroxaban dosage dependent on renal function were recruited in the trial. The objective of this observational research was to compare Anti-Xa levels at peak and trough between the standard dose and Japan-specific dose of rivaroxaban. Patients were treated with standard-dose for at least 1 week and switched to Japan specific-dose for at least 1 week.  A brief detail of the study population and randomization was added under “study design”. (line 107-112)

The analysis included data from a previous study comparing Anti-Xa levels at peak and trough between the standard dose and a Japan-specific dose of rivaroxaban in Thai patients [25]. Sixty adult patients with non-valvular AF were included in the previous study at a tertiary hospital that serves as an academic and referral center for northern Thailand between June 2018 and January 2019. Patients with significant renal impairment (CrCl <15 mL/min) or poor medication adherence were excluded [25].”

In addition, the summary of patient characteristics was added into Table 1:

Table 1. Summary of patient characteristics (n=60)

Characteristics

Value

Male, n (%) 

38 (63.3)

Age (y), mean (SD)

69.4 (9.2)

Body weight (kg), mean (SD)

64.0 (14.1)

Body mass index (kg/m2), median (Q1, Q3)

24.2 (21.5, 26.9)

Creatinine (mg/dL), mean (SD)

1.1 (0.3)

Creatinine clearance (mL/min), mean (SD)

59.0 (22.8)

CHADS2 score, median (Q1, Q3)

2 (1, 2)

CHA2 DS2-VASC score, median (Q1, Q3)

3 (2, 4)

HAS-BLED score, median (Q1, Q3)

2 (1, 2)

Underlying disease, n (%) 

            Hypertension

            Dyslipidemia

            Diabetes

            Congestive heart failure

            Ischemic heart disease

            Ischemic stroke

46 (76.7)

35 (58.3)

16 (26.7)

15 (25.0)

8 (13.3)

7 (11.7)

Concomitant medications, n (%) 

            Dronedarone

            Amiodarone

            Aspirin

            Clopidogrel

3 (5.0)

1 (1.7)

1 (1.7)

1 (1.7)

  1. Did the 15-mg group (Figures 3 and 4) include both patients whose standard dose was reduced because of a low CrCl level and patients with a Japan-specific dose and normal renal function? This should be explained.

Figures 3 and 4 illustrate the steady-state CMAX and CMIN of rivaroxaban simulated with different dosing regimens using the model's final parameters. In the simulations, the important covariates (e.g., CrCl and body weight) were considered and stratified to demonstrate the effect of varying covariate levels on simulated exposures. These numbers may give justification for defining optimal dosage regimens for rivaroxaban. For instance, if 10 mg is provided once daily, the majority of individuals with normal renal function would have a rivaroxaban exposure lower than the typical exposure values. If 20 mg once day is administered, the majority of patients with impaired renal function would have an exposure to rivaroxaban that exceeds the typical exposure range.

  1. Figure 3: please add that the 5th to 95th percentile of rivaroxaban concentrations (dashed lines) was obtained in patients receiving 20 mg rivaroxaban (similar for Figures 4 and 5).

We apologized for this unclear. We have added the statement that the 5th and 95th percentile ranges of rivaroxaban exposure obtained from patients receiving 20 mg rivaroxaban.

(Figure 3; line 270, Figure 4; line 278, Figure 5; line 288)

“Figure 3. Simulated maximum concentration (CMAX) of rivaroxaban with different dosing regimens. The boxplots represent the predicted CMAX stratified by creatinine clearance and body weight. (A) rivaroxaban 10 mg once daily. (B) rivaroxaban 15 mg once daily. (C) rivaroxaban 20 mg once daily. The dashed lines represent the 5th and 95th percentile ranges of the typical CMAX (184–343 ng/mL) reported in patients with non-valvular atrial fibrillation receiving 20 mg of rivaroxaban [19].”

“Figure 4. Simulated minimum concentration (CMIN) of rivaroxaban with different dosing regimens. The boxplots represent the predicted CMIN stratified by creatinine clearance and body weight. (A) rivaroxaban 10 mg once daily. (B) rivaroxaban 15 mg once daily. (C) rivaroxaban 20 mg once daily. The dashed lines represent the 5th and 95th percentile ranges of the typical CMIN (12–137 ng/mL) reported in patients with non-valvular atrial fibrillation receiving 20 mg of rivaroxaban [19].

“Figure 5. Simulated AUC0-24 of rivaroxaban with different dosing regimens. The solid black lines represent the median AUC0-24 The shaded areas represent the 90% prediction interval of the model. The dashed lines represent the 5th and 95th percentile ranges of the typical AUC0-24 (1,860–5,434 ng· h/mL) reported in patients with non-valvular atrial fibrillation receiving 20 mg of rivaroxaban [19].”

  1. Table 2 is difficult to read and therefore not very informative. An attempt should be made to find a graphical representation.

Thank you for your suggestion. We have created a graph from Table 2 data to make it easier to understand. [Figure 6]

Figure 6 Proportion of patients having predicted CMAX, CMIN, and AUC0-24 within the typical exposure ranges reported in patients with non-valvular atrial fibrillation (1000 simulations per subgroup).

  1. The reference list should be changed: Several references are redundant and can be omitted (for example, 35 and 36), some are inappropriate (for example, reference #38 is a review and not an original paper, and reference #39 is inappropriate), and additional references should be included to further explain the role of rivaroxaban concentration in complications. Reference DOI: 10.1177/0091270006296058 should also be included and commented on.

We appreciate your feedback. We have deleted references #35, #36, #38, and #39 and added additional references to clarify the significance of rivaroxaban concentration in complications. [63-68]

“In addition, a previous study showed an elevated risk of thrombotic events in patients with low Anti-Xa level measured at trough concentration [11,12]. Therefore, a precise dosage of rivaroxaban would be required to provide optimum exposure. Furthermore, substantial interpatient variability in the trough concentration of rivaroxaban has been observed, which raises the risk of unexpected bleeding [13,14].”

References

  1. Miklič M, Mavri A, Vene N, Söderblom L, Božič-Mijovski M, Pohanka A, et al. Intra- and inter- individual rivaroxaban concentrations and potential bleeding risk in patients with atrial fibrillation. Eur J Clin Pharmacol. 2019;75(8):1069-75.
  2. Sennesael AL, Larock AS, Douxfils J, Elens L, Stillemans G, Wiesen M, et al. Rivaroxaban plasma levels in patients admitted for bleeding events: insights from a prospective study. Thromb J. 2018;16:28.

The impact of age and body weight on the pharmacokinetics (PK) of rivaroxaban was discussed in detail, and a suggested reference was provided along with comments. [line 344-350]

“These results are consistent with earlier findings showing that the CMAX of rivaroxaban was unaltered in patients with high body mass but increased by 24% in those with severely low body mass (50 kg), resulting in a prolonged prothrombin time [43]. A prior study indicated that elderly patients were exposed to 41 percent more rivaroxaban than young subjects [44]. However, the increasing exposure with age was mostly attributable to a lower clearance as a result of decreased renal function [37,44].”

References

  1. Kubitza D, Becka M, Zuehlsdorf M, Mueck W. Body weight has limited influence on the safety, tolerability, pharmacokinetics, or pharmacodynamics of rivaroxaban (BAY 59-7939) in healthy subjects. J Clin Pharmacol. 2007;47(2):218-26.
  2. Kubitza D, Becka M, Roth A, Mueck W. The influence of age and gender on the pharmacokinetics and pharmacodynamics of rivaroxaban--an oral, direct Factor Xa inhibitor. J Clin Pharmacol. 2013;53(3):249-55.

  1. The discussion should be shortened and portions that are repetitions of the text in the introduction (e.g., lines 285 - 290) should be removed.

This paragraph under the “Discussion” was removed.

  1. Line 246: "Asian patients" are actually Japanese patients. Please be specific.

Thank you for your suggestion. We have revised these words (lines 309).

“Because of differences in patient characteristics, the Japanese patients had greater rivaroxaban concentrations when treated with the approved dosage [23,24].”

  1. I suggest shortening the title to "Population pharmacokinetics and dose optimization of rivaroxaban in Thai patients with non-valvular atrial fibrillation" as it is sufficiently informative.

Thank you for your suggestion. The title has been revised as follows:

"Population pharmacokinetics and dose optimization based on renal function of rivaroxaban in Thai patients with non-valvular atrial fibrillation".

  1. Anti-Xa is the standard abbreviation for the test.

We have revised the abbreviation “Anti-Xa” throughout the manuscript.

  1. DOAC is preferred over NOAC according to ISTH.

The phase " NOAC" was changed to "DOAC" throughout the manuscript.

Round 2

Reviewer 1 Report

The authors have addressed the comments from this reviewer.

Author Response

  1. The laboratory measurement of rivaroxaban was performed by using the chromogenic method. Please discuss how the performance of this method compared to LC-MS/MS assays based on recent articles available in the literature.

Thank you for your suggestions. We have revised “Discussion” to provide the performance of the chromogenic method compared with LC-MS/MS assays.

Line 412-420

“In this investigation, rivaroxaban concentrations were determined indirectly using commercially available Anti-Xa assays (BIOPHENTM Heparin LRT) with validated specific rivaroxaban calibrators. Despite the fact that liquid chromatography with tandem mass spectrometry (LC-MS/MS) is the gold standard approach for direct detection of rivaroxaban plasma concentrations [46], it is time-consuming, technically demanding, and not always available. The technique comparison research revealed a correlation between the BIOPHENTM Heparin LRT and LC-MS/MS test at all rivaroxaban concentrations (r2=0.97) [47]. Thus, anti-Xa tests can be used to detect rivaroxaban levels indirectly in clinical practice.”

References

  1. Gosselin RC, Adcock DM, Bates SM, Douxfils J, Favaloro EJ, Gouin-Thibault I, et al. International Council for Standardization in Haematology (ICSH) Recommendations for Laboratory Measurement of Direct Oral Anticoagulants. Thromb Haemost. 2018;118(3):437-50.
  2. Königsbrügge O, Quehenberger P, Belik S, Weigel G, Seger C, Griesmacher A, et al. Anti-coagulation assessment with prothrombin time and anti-Xa assays in real-world patients on treatment with rivaroxaban. Ann Hematol. 2015;94(9):1463-71.

  1. Please provide a full explanation of why body weight was selected a covariate instead of the body-mass index, along with its advantages/limitations of this choice.

Thank you for highlighting these concerns. In our study, possible factors were explored using a stepwise strategy with forward inclusion and backward deletion. The difference of OFV>3.84 (c2, degree of freedom 1, p<0.05) and OFV >6.63 (c2, degree of freedom 1, p<0.01) was chosen as the cutoff for forward inclusion and deletion, respectively. During covariate analysis, the influence of body-mass index (BMI) was not discovered as a significant covariate for either volume of distribution (V/F) or clearance (CL/F) (p>0.05). Therefore, BMI was not included in the final model. However, body weight was a significant covariate for rivaroxaban's V/F (DOFV= -7.33, p<0.01) with an improvement of the goodness of fit. Consequently, body weight was included in the final model.

We have amended "Discussion" to provide a full explanation for why body weight was chosen as a significant covariate rather than body mass index. Additionally, their benefits and drawbacks have been discussed. (Line 355-365)

“It has been determined that body size (i.e., lean body mass and body surface area) influences the PK characteristics of rivaroxaban [20, 34, 37, 39]. One research involving healthy volunteers demonstrated that BMI influences the V/F of rivaroxaban [38]. Moreover, the results from previous studies showing that CrCl was the significant covariate influencing rivaroxaban CL/F, whereas other body size measurements alone were not found to significantly impact rivaroxaban CL/F [40, 41]. In our study, BMI was not detected as a significant covariate for either V/F or CL/F (p>0.05) during covariate analysis. However, body weight was a significant covariate for V/F (p<0.01) with an improvement of the goodness of fit. In clinical practice, body weight is more readily available than other weight measurements. Thus, using body weight as a measurement of body size would be more practical.”

  1. The paper seems to be controversial on the role of body weight. Once established that it has an impact on V/F, it should be explained why it does not have a clinically relevant impact on the clearance of rivaroxaban. The sentence in lines 352-353 suggests that methodological factors may have prevented the authors from being able to evaluate the role of body weight in a way so that it can be incorporated into the final recommendation for dose selection. Please clarify this.

We apologize for the confusion. In covariate model development, the effect of body weight on pharmacokinetic parameters V/F and CL/F was examined. Our results demonstrated that bodyweight did not reach a significant level (p>0.05) to explain the variability of CL/F. The only significant covariate affecting CL/F or rivaroxaban was the creatinine clearance (CrCl). As body weight was included in the CrCl calculation, it appears that body weight indirectly influences the CL/F of rivaroxaban.

Lines 352-353 stated that as V/F was estimated without its interindividual variability (IIV), the effect of weight in explaining part of the IIV of V/F cannot be identified. However, weight was retained as a significant covariate on V/F in the final model since it significantly decreased the OFV and improved the overall model fit.          

We amended "Discussion" to clarify the impact of body mass on PK parameters of rivaroxaban. In addition, lines 352-353 of the phrase were altered to make them clearer (Line 366-369)

“In our analysis, the IIV of V/F could not be estimated from the data, thus the V/F was estimated without its IIV. Although, the impact of body weight in explaining part of the IIV of V/F cannot be determined, weight was kept in the final model as significant covariate on V/F as it showed significantly decreased of the OFV and improve the overall model fit.”
